# Long-Term Water Absorption of Hybrid Flax Fibre-Reinforced Epoxy Composites with Graphene and Its Influence on Mechanical Properties

**DOI:** 10.3390/polym14173679

**Published:** 2022-09-05

**Authors:** Amer Oun, Allan Manalo, Omar Alajarmeh, Rajab Abousnina, Andreas Gerdes

**Affiliations:** 1Centre for Future Materials, University of Southern Queensland, Toowoomba, QLD 4350, Australia; 2School of Engineering, Faculty of Science and Engineering, Macquarie University, Sydney, NSW 2109, Australia; 3KIT Innovation Hub, Department of Civil Engineering, Geo and Environmental Sciences, Karlsruhe Institute of Technology, 76131 Karlsruhe, Germany

**Keywords:** moisture absorption, flax fibre, natural fibre composite, mechanical properties, graphene nanoparticles, SEM

## Abstract

Interest in the use of natural fibres as an alternative for artificial fibres in polymer composite manufacturing is increasing for various engineering applications. Their suitability for use in outdoor environments should be demonstrated due to their perceived hydrophilic behaviour. This study investigated the water absorption behaviour of hybrid flax fibre-reinforced epoxy composites with 0%, 0.5%, 1% and 1.5% graphene by weight that were immersed in water for 1000, 2000, and 3000 h. The flexural and interlaminar shear strength before and after immersion in water was then evaluated. The results showed that graphene nanoparticles improved the mechanical properties of the composites. The moisture absorption process of hybrid natural fibre composites followed the Fickian law, whereas the addition of graphene significantly reduced the moisture absorption and moisture diffusion, especially for hybrid composites with 1.5% graphene. However, the flexural and ILSS properties of the composites with and without graphene decreased with the increase in the exposure duration. The flexural strength of hybrid composites with 0%, 0.5%, 1% and 1.5% graphene decreased by 32%, 11%, 17.5% and 13.4%, respectively, after exposure for 3000 h. For inter-laminar shear strength at the same conditioning of 3000 h, hybrid composites with 0.5%, 1% and 1.5% graphene also decreased by 13.2%, 21% and 17.5%, respectively, compared to the dry composite’s strength. The specimens with 0.5% graphene showed the lowest reduction in strength for both the flexural and interlaminar tests, due to good filler dispersion in the matrix, but all of them were still higher than that of flax fibre composites. Scanning electron microscope observations showed a reduction in voids in the composite matrix after the introduction of graphene, resulting in reduced moisture absorption and moisture diffusion.

## 1. Introduction

Interest in the use of natural fibres as a replacement for artificial fibres as reinforcements in polymer composites has increased in the last decade due to their advantages, such as being renewable resources and their abundance, recyclability, biodegradability, low density, good mechanical characteristics, light weight and low cost [1,2]. Due to these advantages, natural fibres are now utilised in building and construction, automotive parts and other industrial applications [3,4,5]. However, natural fibres are found to be highly sensitive to moisture and are incompatible with most polymer matrices [6,7]. These issues are related to their hydrophilic nature, where they tend to absorb a high amount of moisture when immersed in water or exposed to an extremely humid environment. As a result of moisture absorption, natural fibres used as reinforcement swell and cause micro-cracks in the polymer matrix, which directly affect most composite properties [8,9]. Therefore, the water absorption of natural fibre composites should be evaluated for their wide use in different engineering applications [10].

The sensitivity of natural fibre-reinforced polymer (NFRP) composites to humid environmental conditions has been highlighted by several researchers [11,12,13]. These researchers concluded that the exposure of composites to an aggressive environment for a long period of time caused their mechanical properties to be reduced. The variation in the mechanical strength reduction depends on the type of fibre used as reinforcement, the filler used as an additive and the resin used as a matrix. A number of critical parameters govern the moisture diffusion behaviour of the NFRP composites [14,15,16] including the diffusion of water molecules through micro-cracks in the polymer matrix composites. Other mechanisms are the capillary transport of water molecules along interfacial bonding between the fibre and the polymer matrix and the water diffusion in the micro-cracks of the polymer matrix caused by fibre swelling. It should be highlighted that the swelling of natural fibres constrains their use as effective internal reinforcement for polymeric composites in outdoor applications as it reduces the long-term service of these composites under wet conditions by weakening the fibre/matrix interface. An understanding of the diffusion behaviour of NFRP composites is therefore necessary.

There are a number of factors contributing to the water absorption of plant fibre, including the porosity and internal structure of the fibre [17,18]. On the other hand, the cellulose content in the plant fibre reduces water absorption and makes the fibre structure more hydrophilic. Among all available plant fibres, flax fibre has one of the highest cellulose contents, reaching approximately 72% [19,20], as shown in Table 1, and a low micro fibril angle of 2 to 8 degrees. The higher degree of cellulose content in flax fibres is responsible for the strength of the fibres, while the latter property controls the stiffness. Both contribute to the mechanical properties of the fibre and the performance of the produced composites [21]. The cellulose contains hydroxyl groups, which interact with water molecules to form a hydrogen bond. Due to this hydrophilic character, flax fibres become less compatible with hydrophobic polymer matrices [16]. Although this inherent incompatibility has been treated with physical and chemical techniques such as alkali, peroxide and silane treatment, NFRP composites still suffer from this issue. More efficient techniques should be explored to enhance the bonding of natural fibres and the polymer matrix and to improve their moisture absorption properties.

Recently, nanomaterials such as titanium dioxide (TiO_2_), carbon nanotubes (CNTs) and multiwalled nanotubes (MWNT) as secondary reinforcement have been drawing the attention of researchers due to their excellent properties when incorporated into polymer matrices. Their engagement improves the physical, mechanical and thermal properties of neat resin and polymer composites. [22] reported that the hybridization of natural fibre (5, 10 and 20 wt%) with 2% and 5% of montmorillonite (MMT) in styrene-butadiene-styrene triblock copolymer matrix composites showed significant improvement in the tensile strength, tensile modulus, and elongation at break for composites filled with 2% MMT and reinforced with 5% by weight of natural fibre. In addition, the water absorption behaviour of the composites filled with 2% MMT and reinforced with 5% by weight of fibres decreased by approximately 15% after 400 h of immersion in water. A study by Nayak et al. [23] observed improved mechanical properties of glass fibre-reinforced epoxy hybrid composites with the inclusion of 0.1%, 0.3% and 0.7% of TiO_2_ and immersed in water for 625 h. The flexural strength retention and ILSS strength retention were improved by 19% and 18%, respectively, for hybrid composites with 0.1% of TiO_2_, while also reducing the moisture absorption by 9%. On the other hand, the hybrid composites with 0.7% of TiO_2_ showed a maximum improvement in modulus retention of 22% compared to the glass fibre composites. They attributed the improvement in mechanical properties and water absorption behaviour to the addition of nano TiO_2_. The effective aspect ratio of TiO_2_ particles improves the interface adhesion by creating additional sites of mechanical crosslinking, thereby improving the properties. Moreover, Kushwaha et al. (2014) found that the inclusion of 0.15% by weight of carbon nanotubes to bamboo fibre-reinforced epoxy composites improved the mechanical and moisture absorption properties by 6.7%, 2.7%, 5.8%, 31%, 85% and 11.8%, respectively, for tensile strength, tensile modulus, flexural strength, flexural modulus, impact strength and moisture absorption after 1600 h of conditioning in water compared to a composite without carbon nanotubes. The improvement in the mechanical properties and reduced water absorption with the addition of carbon nanotubes is due to increasing the bonding strength at the interface, which leads to a high-performance composite. These researchers concluded that the addition of nanofiller to the polymer matrix plays a significant role in minimizing water absorption by reducing free volume spaces in terms of the amount, size and distribution of these spaces, depending on the uniform dispersion of the nano-filler in the polymer matrix. They also mentioned that the effect of adding nano-filler improved the interface area by making strong covalent bonds between this filler and the polymer matrix, thus attempting to restrict the ability of water molecules to move freely in the interface area. In addition to that, the barrier properties of nano-filler can contribute to the formation of a zigzag path that slows the amount of water diffusion through the composite matrix.

The mechanical performance of NFRP composites in dry conditions is considered acceptable, but the information on the behaviour of these composites under long-term wet environmental conditions is limited. Thus, further investigation is required to fully understand the effects of graphene nanoparticles on the durability performance of natural fibre composites in a humid service environment as this performance is considered a critical factor for safety standards for using this material for outdoor applications. This study investigated, for the first time, the effects of graphene nanoparticles on the water absorption behaviour of natural flax fibre composites to extend their use in long-term wet conditions and to understand the critical parameters governing moisture diffusion behaviour.

The experimental works focused on the water absorption behaviour of flax-natural fibre composites with graphene at different percentages by weight (0%, 0.5%, 1% and 1.5%) and immersed in water for 1000, 2000 and 3000 h. The effects of these parameters on the flexural and inter-laminar properties before and after conditioning were evaluated. The results of this study will provide a better understanding of the behaviour of flax-natural fibre composites filled with graphene nanoparticles under wet conditions, which is crucial for their wide acceptance and use in external mechanical and structural engineering applications.

## 2. Experimental Program

### 2.1. Materials

Epoxy resin (R246TX) and hardener (H160) (ATL Composites, Queensland, Australia) were mixed and used as a matrix to manufacture the composites. The resin-to-hardener ratio used was 1:4 by weight, as recommended by the supplier. Graphene nanoparticles with an average surface area of 300 m^2^/g were supplied by Sigma-Aldrich, Bayswater, Australia. The unidirectional flax fibres with a thickness of 0.36 mm were supplied by Colan Composite Reinforcement, Huntingwood, Australia. Table 1 shows the internal structure of flax fibres. The properties of the neat epoxy resin, natural fibres and graphene listed in Table 2 are based on the available literature and as described by the manufacturer.

### 2.2. Specimen Fabrication

Figure 1 shows the manufacturing process of the fibre-reinforced epoxy composites. Unidirectional flax fibres were cut into dimensions of 600 mm in length and 400 mm in width (see Figure 1a). A total of six layers were manufactured to obtain a nominal plate thickness of 4 mm. Prior to adding the resin, the flax fibre sheets were placed in an oven at 40 °C for 30 min to remove moisture as per the manufacturer’s recommendations as it would affect the wettability and bond between the resin and fibres. Epoxy resin was mixed for five minutes and poured onto the fibre sheets until saturated using the hand layup technique. The flax fibre layers were then placed on top of each other to build the composites with a fibre volume ratio (V*_f_*) of 25%. The V*_f_* was calculated by weight. A metal roller was used to ensure that the resin was evenly distributed throughout the fibre sheets and to remove any air bubbles while wetting the fibres (see Figure 1b).

Different percentages of graphene nanoparticles by weight (0.5%, 1.0% and 1.5%) were mixed with the epoxy resin before adding the hardener to produce the hybrid composites [24]. It is worth mentioning that 1.5% graphene content was selected due to the high viscose epoxy mixture for higher graphene content. Thus, hard mixing of the mixture was performed using the normal shear mixer beyond 1.5% of graphene, which contributes significantly to form agglomeration. Moreover, it was found by trials and previous research [1,2] that there is no significant enhancement in the mechanical properties beyond 1.5%.

A homogeneous mixture was obtained by using an electric shear mixer for five minutes as per the recommendations of the graphene supplier. After wetting, the sheets were inserted into a vacuum bag sealed with yellow sealant tape to create an airtight seal (see Figure 1c). A constant pressure of 92 kPa was then applied, and the samples were initially left for 24 h to cure (see Figure 1d). Subsequently, the manufactured plates were demoulded and post-cured for 3 h at 120 °C as recommended by the supplier. The composite plates were then cut according to specimen dimensions using a waterjet to produce the test samples (see Figure 1e).

### 2.3. Water Absorption Test

The water absorption and water diffusion coefficient of flax fibre composites and hybrid nanocomposites were measured following ASTM D570 [25]. Initially, all samples were weighed in a dry condition after coating their edges with a thin layer of resin to ensure the entry of moisture is only through the top and bottom surfaces of the composites, as was also implemented by [26]. The samples were then immersed in tap water for 1000, 2000 and 3000 h. After 24 h of immersion, the samples were taken out of the water, dried with tissue paper, and immediately weighed with a digital scale of 0.001 mg accuracy. Once the samples were weighed, they were immediately re-immersed in water. At regular intervals, the weighting process was repeated over 126 days of immersion in water. Moisture absorption was then calculated as the weight difference between dry and wet samples, and the moisture content percentage (*M_t_%*) was calculated using Equation (1).
(1)$$M_{t}\left(\%\right)=\left(\frac{W_{t}-W_{0}}{W_{0}}\right)\times 100$$
where *W*_0_ and *W_t_* are the weight of the dry and wet samples after time *t*, respectively.

On the other hand, Equation (2) was used to calculate the diffusion coefficient (*D*) assuming that the moisture diffusion behaviour of composites follows the Fickian diffusion behaviour as shown in Figure 2.
(2)$$D=\pi {\left(\frac{Kh}{4Mm}\right)}^{2}$$
where *h, K* and *Mm* are the sample thickness, the initial slope of the plot of *M(t)* against *t^½^* and the maximum increase in weight, respectively.

### 2.4. Mechanical Testing

The mechanical properties of the controlled and conditioned hybrid composites were evaluated under bending and interlaminar shear. Both tests were performed using the 10 kN MTS machine under a 3-point test setup at a loading rate of 1.3 mm/min (see Figure 3a) but with different span-to-depth ratios. The flexural test samples were 80 mm long, 16 mm wide and 4 mm thick and were tested over a 64 mm support span producing a span-to-depth ratio of 16, as suggested by [27] and shown in Figure 3b. On the other hand, the inter-laminar shear strength (ILSS) test shown in Figure 3c was implemented using 24 mm long by 16 mm wide and 4 mm thick test specimens, and with a support span of 20 mm producing a span-to-depth ratio of 5 as suggested by ASTM D2344 [28].

### 2.5. SEM Observations

The scanning electron microscope (SEM) JEOL JXA 840A (Jeol, Tokyo, Japan) was used to examine the damage features, fracture surface and fibre–matrix interface of the dry and wet samples with and without graphene after prolonged immersion in water and tested at room temperature. The graphene distribution within the composite matrix was also evaluated by SEM, after carefully preparing the samples with dimensions of 10 mm by 10 mm from the tested specimens.

## 3. Results and Discussion

### 3.1. Dynamic Mechanical Analysis (DMA)

The thermo-mechanical properties of the polymer have been evaluated by using a DMA device, a Q800 type (Mettler-Toledo Ltd, Melbourne, Australia) of thermal analysis instrument, to determine the glass transition temperature of dry and wet samples based on [29]. DMA test samples were cut with dimensions of 50 mm × 8 mm × 4 mm. After clamping the specimens in a dual-cantilever system, DMA tests were performed using a multi-frequency strain mode, with a temperature increase rate of 5 °C/min to scan the temperature from room temperature to 120 °C.

Figure 4 shows the dynamic mechanical analysis (DMA) curves of hybrid composites with different graphene loadings after immersion in water for 3000 h. The temperature at the highest peak value of the storage modulus was considered to be the glass transition temperature (Tg) of the sample. The DMA results in Table 3 showed that exposure of these composites to water caused a lower storage modulus as well as the glass transition temperature (Tg) values compared with dry samples, as was also reported by Saha et al. [30] for thermo-mechanical properties of flax-hemp/epoxy hybrid composites. The main reason for lower values of the Tg is the plasticization of the epoxy matrix caused by water absorption, which normally acts as a plasticizer [31]. Similarly, Al Rifai et al. [32] observed lower glass transition values induced by the plasticization effect for the long-term durability of basalt fibre-reinforced polymer bars. Moreover, the trend of Tg results in this study is in agreement with other studies reported in the literature such as [33]. On the other hand, it can be noted from Figure 4 that the wet composites with 1% and 1.5% graphene showed a higher storage modulus than the dry samples. Qin et al. [34] highlighted that this phenomenon can be explained by the better orientation of the molecular chain for wet samples as of the presence of moisture. Thus, higher-oriented chains result in better shear resistance, and higher potential energy in the wet specimens revealed higher storage energy than the dry samples. This study also showed that adding nanofillers enhanced the molecular chain, resulting in an obvious enhancement in the storage modulus. Further studies are recommended to investigate this aspect.

The effect of the plasticization on Tg can be explained by the weak interactions between the epoxy resin and graphene nanoparticles. This means that the weak filler/matrix interaction causes the glass transition temperature to drop as pointed out by [35]. With the increase in the filler content, agglomeration of the filler occurs due to the increase in viscosity, which leads to weak bonding by forming micropores between the graphene nanoparticles, allowing the entry of water molecules between these nanoparticles. In this case, the filler agglomerate is joined by weak Van der Waals forces. This can lead to a weakening of the bonding strength of the hybrid composites resulting in a significant drop in Tg values as shown in the samples with 1% and 1.5%.

### 3.2. Moisture Absorption Behaviour of Hybrid Flax Fibre Composites

Figure 5 shows the relationship between the percentage weight gain and the square root of the immersion time for flax/epoxy hybrid composites with different levels of graphene by the weight of the matrix at room temperature. It was found that the moisture absorption process of these specimens follows Fickian law, which can be described as linear at the initial stage of the water absorption curve and then slows down until it approaches the saturation level after a long period of time. The results show that adding graphene nanoparticles to the epoxy matrix results in lower moisture absorption of hybrid flax fibre-reinforced epoxy composites than those specimens without graphene for the three different immersion durations. This is attributed to the presence of graphene nanoparticles, which act as barriers against moisture in the epoxy matrix. The results also showed that the initial moisture absorption rate and the maximum moisture content values decrease as the amount of graphene nanoparticles increases. This is due to the effective aspect ratio of graphene, which created the tortuosity path and improved the bonding strength at the interface. This is supported by the reduction in moisture diffusion coefficients as shown in Table 4. The moisture absorption behaviour results obtained in this study are in agreement with the results reported by [36] for luffa/epoxy composites with graphene. However, the samples with 1% graphene recorded higher water absorption than those with 0.5% graphene. This is owing to more voids being formed in the modified matrix in the former samples, as revealed by the SEM image in Figure 10b. Moreover, Figure 11b shows the presence of small cracks in the interface between the fibre and matrix, which contributes to the penetration of water through the bulk matrix during the immersion process. Zhang and Mi [37] have attributed the development of these inevitable cracks and voids within the composite matrix to the manufacturing and solidification process.

For three different immersion durations, flax fibre epoxy composite samples show poor moisture resistance due to the hydrophilic nature of flax fibre, which is attributed to the hydroxyl groups on its surface. This also explains the higher saturation level of the control specimens than that of the hybrid samples. This phenomenon can be explained by considering the moisture absorption properties of flax fibre. Given the high cellulose content in flax fibre (approximately 72%), the composite absorbs a high amount of water which causes the fibre to swell. This swelling in the flax fibre creates micro-cracks in the epoxy matrix composite. As such, the micro-cracks provided an active capillary mechanism for the water molecules to penetrate into the composite interface through these micro-channels. This inevitably results in swelling stress, which causes deterioration of the natural fibre-based composite properties by separating the flax fibre from the epoxy resin matrix as observed from the SEM. Plasticization due to water absorption is also another factor affecting the performance of the composite fibre/matrix interface, which reduces the bonding strength between them.

Even though flax fibre epoxy composites tend to absorb high amounts of water, graphene nanoparticles seem to limit this tendency by reducing the voids and filling micro-channels, which slows down the diffusion of water molecules into the composite matrix. As a result, the hybrid composites show lower water absorption than those without graphene because the graphene nanoparticles have moisture resistance due to their hydrophobic nature. These nanoparticles attached to the fibre surface act as external protective materials that contribute to reducing the water absorption rate and improving the adhesion of the fibre/resin matrix interface by increasing the surface roughness of the fibre. In the shortest water immersion period (1000 h), the addition of 0.5% graphene nanoparticles decreased both the moisture diffusion coefficient and moisture absorption of flax fibre composites by 41.2% and 68.7%, respectively. In contrast, when these samples were immersed for 3000 h, the moisture diffusion coefficient and moisture absorption content were significantly enhanced by 23.7% and 66.4%, respectively, compared to that of the control samples. It was noted that the increased moisture content and diffusion coefficient with increasing exposure time are due to the internal plasticization of the matrix as evidenced by the reduction in Tg found by dynamic mechanical analysis (DMA) as shown in Table 3. From Table 4, it can be noted that the percentage of moisture absorption increased slightly with an increasing exposure duration for all composites. The moisture absorption results of this study are in agreement with the experimental results from Chaharmahali et al. [38] for hybrid bagasse fibre-reinforced polypropylene composites with graphene. Nevertheless, at any given duration of water immersion, the moisture diffusion coefficient, as well as the moisture absorption content, can be reduced by adding graphene nanoparticles. Chaharmahali et al. [38] also mentioned that the trend of water absorption in bagasse fibre composites decreased with the presence of graphene. These reductions also emphasize the importance of graphene nanoparticles. Graphene nanoparticles have demonstrated their ability to reduce moisture absorption and the diffusion rate by hindering interface plasticization and protecting fibres from swelling through the following mechanisms: (i) A tortuous path is formed by graphene particles, which acts as a longer path to the diffusion of water in the composite matrix, and (ii) providing additional sites for mechanical interlocking between the fibres and the resin through the effective aspect ratio of these nanoparticles attached to a fibre surface, which can improve the bonding performance. Both contribute to the reduction in moisture absorption and the diffusion rate.

### 3.3. Flexural Behaviour of Hybrid Flax Fibre Composites

#### 3.3.1. Effect on Flexural Strength

Figure 6 shows the flexural strength (FS) and strength retention of hybrid flax fibre-reinforced epoxy composites with different graphene percentages in dry and wet conditions. In dry conditions, the FS of the flax fibre composites increased with the addition of graphene. This improvement is due to the graphene providing a better stress transfer mechanism at the interface between the epoxy matrix and the flax fibres. Another simple explanation is attributed to the high surface area of graphene providing an effective chemical interaction with the matrix resin, thus increasing the strength. It should be noted that the average FS value increased, with the highest value recorded for the composite samples with 0.5% due to the good dispersion of nanoparticles. Ashok et al. (2020) and Ashok & Kalaichelvan [36] also stated that the improvement in the flexural, impact and tensile strength of the luffa fibre composite with nano-filler is attributed to the enhancement of the fibre/matrix interface adhesion after nano-filler hybridization, causing an effective load transfer mechanism from the epoxy matrix to the luffa fibres and thus increasing mechanical properties. However, adding more graphene than 0.5% to the epoxy matrix results in lower flexural strength due to the agglomeration of graphene nanoparticles caused by the increase in viscosity of the epoxy resin as shown from the SEM observations (see Figure 11b,c). As such, similar behaviour in increasing viscosity with a high amount of graphene has been confirmed by other studies [39]. The same trend of flexural strength results has been observed by Nayak et al. [23] for the glass fibre-reinforced epoxy composites with nano TiO_2_ particles, which showed a reduction in flexural strength due to the agglomeration effect.

The absorbed water reduced the flexural strength of hybrid composites, as also shown in Figure 6. The FS also decreases with exposure duration. This reduction in flexural strength is associated with an increase in the moisture absorption rate, which results in the plasticization of the epoxy matrix and fibre microstructure, thus impairing the adhesion of the fibres to the epoxy matrix at the interface. This observation is also supported by Table 1, which showed an increase in moisture absorption values with increasing exposure time. Regardless of the exposure duration, the flexural strength of all samples with different graphene percentages decreased after immersion in water when compared to their dry strength, indicating that the properties of the fibre/matrix interface, which govern the flexural strength, are affected by moisture absorption and diffusion. As can be observed in Figure 6, the FS of composites with 0.5% graphene and immersed in water for 1000, 2000 and 3000 h decreased by 3.0%, 7.3% and 13.4%, respectively, compared to the dry composite strength, but is higher than the wet strength of the other specimens. This can be explained by the uniform dispersion of graphene nanoparticles. This uniform dispersion of graphene nanoparticles throughout the matrix limits the available voids in this matrix for water molecules and reduces the diffusion of these water molecules at the composite interface by filling the micro-channels, resulting in higher wet strength. The retention of FS at different immersion durations proves that the addition of graphene increased the bonding strength at the fibre/matrix interface, and reduced moisture absorption, resulting in an increase in the strength of the hybrid composites. However, the composites with 1% graphene exhibited lower FS than those with 0.5% and 1.5%, regardless of the exposure duration, due to higher moisture absorption as a result of the increased number of voids formed in the modified matrix during the fabrication process as well as the presence of small cracks in the sample interface, as witnessed in the SEM (Figure 12b). The slightly lower strength of the wet samples with 1.5% graphene compared with the dry specimens is due to the lower moisture absorption rate. A possible justification for this observation could be the fibre swelling induced by moisture absorption causing pressure on the surrounding matrix. This swelling effect bridged the gap between the fibre and the matrix, thus providing increased adhesion of the fibres to the matrix, resulting in a slight reduction in FS. Muñoz and García-Manrique [16] also observed the positive effect of fibre swelling on the short-term flexural and tensile properties of bio-epoxy composites reinforced with flax fibres. Another possible justification may be due to the additional reduction in free volume in the matrix with this number of graphene particles. The influence of graphene content on flexural strength retention is illustrated in Figure 6b. Generally, flexural strength retention is improved by increasing the amount of graphene for all conditions, compared with specimens without graphene (see Figure 6b). Hybrid composites with 1.5% graphene retained 98%, 95% and 92% of the dry strength for 1000, 2000 and 3000 h, respectively, which was the highest improvement in flexural strength retention. This improvement is due to the reduced moisture absorption within the composite matrix as the graphene particles fill the free spaces in the matrix.

Figure 7 shows the failure mode of hybrid flax fibre-reinforced epoxy composites with graphene at different weight percentages under flexure in wet conditions. From Figure 7a, it can be noted that weak fibre/matrix adhesion at the composite interface without graphene led to flexural crack propagation along the sample interface, which resulted in lower flexural strength. In contrast, the failure mode in flexural test samples with graphene under the same conditions changed from a crack along the fibre/matrix interface to a crack along the thickness of the sample. This indicates the excellent adhesion between the filler/fibre and the matrix, which delays crack initiation and propagation and causes increased flexural strength, as shown in Figure 7b.

#### 3.3.2. Effect on Flexural Modulus

Figure 8 shows the flexural modulus (FM) and modulus retention of flax fibre-reinforced hybrid epoxy composites with graphene at different weight percentages (0%, 0.5%, 1% and 1.5%). In a dry environment, an expected increase in the flexural modulus with increasing graphene was noted because of the high modulus of graphene nanoparticles. Wang and Drzal [40] also mentioned that graphene particles are stiffer materials than epoxy resin, which increases the stiffness of the matrix but makes it more brittle. For wet composites, a reduction in the FM was observed with increasing exposure duration, regardless of the amount of graphene. This reduction in wet modulus can be explained by the amount of moisture absorption, which resulted in plasticization at the interface between the flax fibre and the epoxy matrix. The penetration of water molecules into the composite matrix acts as a plasticizer that facilitates the weakening of the bonding at the interface between the fibres and the synthetic matrix, resulting in a lower modulus. The specimens with 1.5% graphene showed the highest values of flexural modulus at 9.1, 8.7 and 8.4 GPa for 1000, 2000 and 3000 h, respectively, which are 45.7%, 52.3% and 55.8% higher than the other composites without graphene. The reduced moisture absorption in specimens with 1.5% graphene is attributed to the presence of a larger number of graphene particles, which increases the stiffness of the matrix. This increase in matrix stiffness in turn limits the fibre extension in the matrix and thus prevents microcracks in the matrix as supported by the SEM image in Figure 12c. However, a remarkable reduction in FM was observed in wet-modulus specimens with 1% graphene, regardless of exposure time, which was attributed to higher moisture absorption causing the flax fibres to swell. This swelling caused micro-cracks to develop at the interface of the composites (see Figure 12b), which resulted in moisture being absorbed throughout the composite matrix, thus reducing the FM. The increase in exposure duration up to 3000 h results in further deterioration at the interface of the hybrid composite, with 1% graphene resulting in an increased strength loss, as evidenced by the modulus retention shown in Figure 8b. The low modulus retention of wet samples with 1% graphene can be directly attributed to the formation of microcracks in the epoxy matrix and gaps between the fibre and matrix due to the swelling of fibre caused by moisture absorption, which reduced the adhesion at the interface (see Figure 12b). In comparison with dry specimens, the FM of the wet samples with 1.5% graphene decreased by 8.9%, 12.8% and 16.3%, respectively, for 1000, 2000 and 3000 h of water immersion. It can be assumed that moisture absorption affected the flax fibre, and hence, FM is considered to be more sensitive to the fibre properties [15]. Another reason for this modulus reduction is due to the degradation of the fibre/matrix adhesion. Regardless of the exposure duration and amount of graphene, all specimens exhibited higher flexural modulus than the samples without graphene. This further confirms that graphene nanoparticles can decrease the free volume in the epoxy matrix and increase the contact area and chemical interactions with the epoxy matrix due to their higher surface area, resulting in reduced moisture absorption and distribution. It can be concluded from these observations that the inclusion of graphene nanoparticles in the epoxy matrix helped reduce moisture absorption and diffusion at the interface of natural fibre composites. as well as improve the bonding strength, thus increasing the performance of these composites. Figure 8b indicates the influence of graphene addition on FM retention under different immersion durations. Hybrid composites exhibited higher modulus retention at 0.5% and 1.5% graphene than those without graphene in all conditions. However, specimens with 1% graphene retained only 64%, 58% and 51% of their dry modulus for 1000, 2000 and 3000 h, respectively, which is lower than the modulus retention of the control composite tested under the same conditions. This reduction in modulus retention is attributed to higher moisture absorption of these samples.

### 3.4. ILSS Behaviour of Hybrid Flax Fibre Composites

Figure 9 shows the inter-laminar shear strength (ILSS) of the dry and wet hybrid composites. Regardless of the exposure duration, similar ILSS behaviour can be observed for wet specimens with different amounts of graphene. Wet composites with 0.5% graphene failed at the highest ILSS strength of 15.9 MPa for 1000 h, which is 11% lower than the dry composites. This is the same mechanism observed in flexural strength, where there is a good filler distribution in the matrix. However, the highest reduction in wet ILSS strength was observed for specimens with 1% graphene, which was 20%, 23% and 25% lower than that of the dry ILSS strength for 1000, 2000 and 3000 h, respectively, as a result of the plasticization effect at the flax fibre/epoxy matrix interface from higher moisture absorption, which has been demonstrated at lower Tg values. This effect could be more pronounced by the agglomeration of graphene nanoparticles as the epoxy becomes more viscous after increasing the amount of graphene (see Figure 12b), thus the interfacial stress concentration causes crack initiation and propagation, as also observed by Kong et al. [41]. This observation could be the significantly reduced effective surface area at 1% graphene loading, which suffers from non-uniform dispersion of the graphene or poor distribution during mixing. Loste, Loste et al. [42] and Neitzel et al. [43] mentioned that nanoparticles naturally show a strong aggregation tendency, which is attributed to the strong attraction between the particles. This aggregation causes a lower effective aspect ratio of these particles, which reduces the interaction between the fibre and the matrix, thus reducing adhesion at the fibre/matrix interface. Flax fibre swelling is another reason to reduce the strength of these samples by causing microcracks at the interface of these specimens and thus increasing the moisture content as discussed beforehand. The SEM observations in Figure 12 support these justifications. However, increasing the amount of graphene by weight to 1.5% in the epoxy matrix showed a reduction in water permeability through the hybrid flax composite interface, which can be attributed to the barrier mechanism generated by these nanoparticles, showing slightly lower ILSS strength. For comparison purposes, the ILSS strength of the wet samples regardless of the exposure duration and graphene content is lower than that of the dry composites because of the deterioration of the flax fibre/epoxy matrix interface adhesion as a result of the moisture-induced plasticization effect. Alaaeddin et al. [44] have also attributed the low mechanical properties of natural fibre composites filled with nanofillers immersed in water to the formation of hydrogen bonds through chemical interactions of water molecules with hydroxyl groups of cellulose, thus impairing the interfacial adhesion of the reinforcements to the matrix. The ILSS retention results presented in Figure 9b show that the addition of graphene can significantly contribute to the retained strength of 89%, 84% and 82% for 0.5% graphene, 84%, 81% and 79% for 1% graphene, and 92%, 88% and 87% for 1.5% graphene at 1000, 2000 and 3000 h, respectively. The control composites retained 83%, 80% and 78%, respectively, under the same conditions. The maximum improvement in ILSS retention was observed for specimens with 1.5% in all conditions due to the low amount of moisture absorption.

Figure 10 shows the inter-laminar shear failure of hybrid flax-reinforced epoxy-based composites. As shown in Figure 10a, specimens without the graphene addition failed due to the inter-laminar shear between the layers of the fibres along the interface due to weakness at the fibre/matrix interface. The specimens with graphene, however, show the same failure behaviour before and after exposure to water. Tensile failure rather than inter-laminar shear mode was observed in hybrid flax composites regardless of the graphene weight ratio, which was attributed to the stronger fibre/matrix interface developed by adding nanoparticles (see Figure 10b). This may also be due to the epoxy matrix becoming stronger and stiffer than the flax fibres due to the addition of graphene.

**Figure 10 polymers-14-03679-f010:**
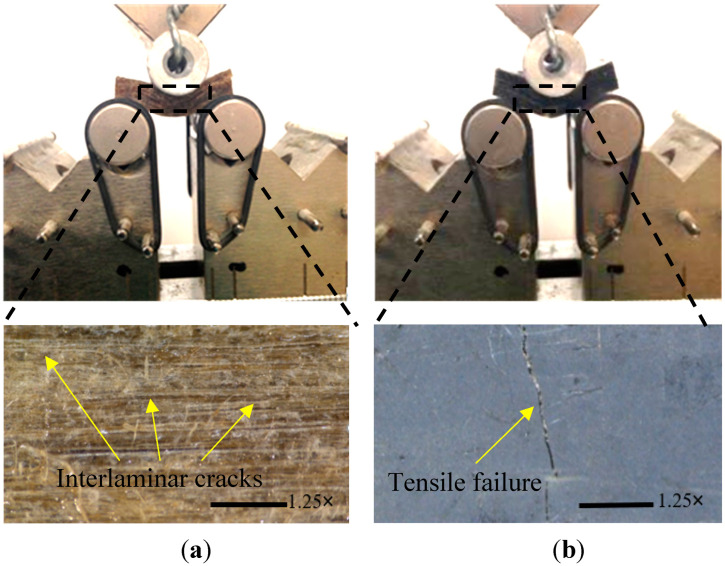
ILSS failure of hybrid composites under wet conditions: (**a**) Composite without graphene and (**b**) composite with graphene.

### 3.5. SEM Image Observations and Analysis

Figure 11 and Figure 12 show SEM images of the microstructure of hybrid composites under wet and dry conditions at room temperature. These images (up to ×500 magnification) show the internal structure of the hybrid composites in terms of the dispersion of graphene particles in the matrix. Regardless of exposure time, adding 0.5% graphene nanoparticles to the epoxy matrix in dry composites improved the fibre–matrix interface resulting in more fibre fractures rather than fibre pull-out (see Figure 11a). However, adding more graphene results in aggregation due to its high surface area and van der Waals forces between nanoparticles, and agglomeration because of the increased viscosity of the epoxy resin. This then causes an increase in the number of void formations in the hybrid composites (see Figure 11b,c). This aggregation/agglomeration results in an imperfect adhesion of the fibre/matrix interface by creating gaps between the flax fibre and the synthetic matrix, which reduces mechanical performance, as observed by Ashok & Kalaichelvan [36] for luffa fibre-reinforced epoxy composites with graphene. In contrast, after immersion in water, the wet specimens with 0.5% graphene exhibited excellent bonding strength between the flax fibre/filler and the resin matrix caused by the uniform dispersion of graphene nanoparticles, which reduced the number of voids in the matrix as revealed in Figure 12a. Figure 12b shows that the deterioration in fibre/matrix interfacial adhesion and matrix cracking in the wet composites at 1% were characterised by the appearance of a gap between the flax fibre and the epoxy matrix induced by the plasticization of the matrix. This indicates weak adhesion of the fibre to the matrix at the interface. For specimens with 1.5%, no fibre pull-out can be observed due to fibre swelling induced by moisture absorption causing pressure on the surrounding matrix, and the effective aspect ratio of graphene also increased the adhesion of the resin to the fibre despite the occurrence of filler agglomeration. This also explains the lowest moisture absorption in these specimens.

**Figure 11 polymers-14-03679-f011:**
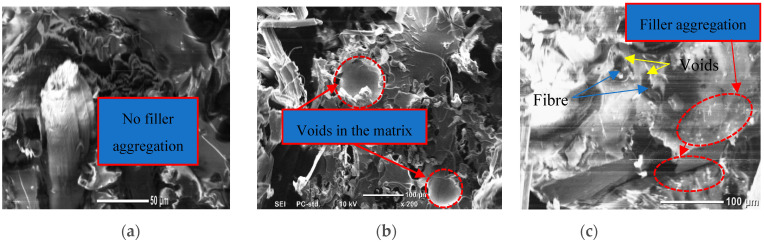
SEM images of dry hybrid composites at different graphene weight ratios: (**a**) Filled with 0.5%; (**b**) filled with 1% and (**c**) filled with 1.5%.

**Figure 12 polymers-14-03679-f012:**
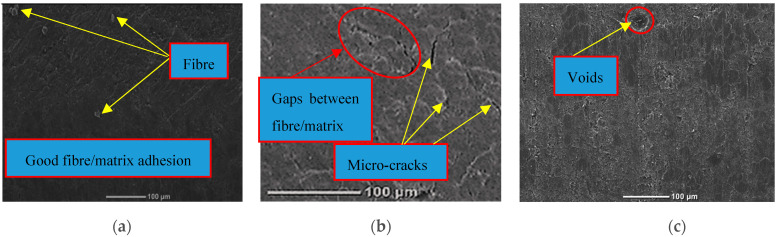
SEM images of wet hybrid composites at different graphene weight ratios: (**a**) Filled with 0.5%; (**b**) filled with 1% and (**c**) filled with 1.5%.

## 4. Conclusions

This study investigated the effect of moisture on the flexural and inter-laminar shear properties of hybrid flax-fibre-reinforced epoxy composites. Hybrid composites with graphene content of 0, 0.5, 1.0 and 1.5% by weight of the epoxy resin were prepared and immersed in water for 1000, 2000 and 3000 h at room temperature. From the test results and observations, the following conclusions can be made:

The addition of graphene nanoparticles decreased the moisture absorption and moisture diffusion coefficient of flax-fibre-reinforced epoxy composites due to the graphene providing a barrier and a tortuous path to the matrix. Hybrid composites with 1.5% graphene exhibited the lowest moisture absorption rates and diffusion coefficients, which were 73%, 72%, and 71% and 71%, 49% and 26% lower, respectively, for 1000, 2000 and 3000 h than the ones without graphene.Graphene nanoparticles have a beneficial effect on the flexural and inter-laminar properties of flax fibre epoxy composites under wet conditions. This effect is mainly due to the improvement in the interfacial adhesion of the flax fibre to the epoxy matrix, and the increase in the stiffness and strength of the epoxy matrix is due to the addition of high-stiffness graphene.The moisture absorption affected FS and ILSS regardless of the graphene weight ratio, where a continuous decreasing trend was observed with increasing exposure durations. This is due to the matrix properties governing these two properties at room temperature.Longer-exposure duration deteriorates the interface of the hybrid composites more than shorter exposure times due to the increased ingress of water affecting its mechanical strength. Immersing the specimen with 0.5% graphene in water for 3000 h showed 10.7% and 7.3% higher reduction in flexural and ILSS strength, respectively, as compared to the same sample immersed in water for 1000 h.Regardless of the graphene weight ratio, the flexural modulus is the most significantly affected mechanical property by moisture absorption as this property is governed by fibre properties. Increasing the immersion time increased the water absorbed by the flax fibres and further reduced the flexural modulus as the absorbed water caused continuous deterioration of the flax fibre.Wet conditioning changed the failure behaviour of hybrid flax fibre composites. The laminates without graphene failed due to weak interfacial adhesion between the fibre and the matrix as evidenced by the crack propagation behaviour along the interface, whereas the hybrid composites with graphene failed by the flexural crack showing strong bonding strength at the interface. Under ILSS, the composites without graphene failed in inter-laminar shear mode originating either from the middle or ends of the samples, but those with graphene showed flexural failure due to the stronger fibre/matrix interface.

The overall findings from this work highlighted the benefits of graphene in decreasing the moisture absorption of flax fibre epoxy composites and enhancing their strength retention, which will enable this type of natural fibre composite for use in wet environments. Further studies can, however, be conducted to investigate the moisture absorption of hybrid flax fibre epoxy composites longer than the exposure time considered in this study.

## Figures and Tables

**Figure 1 polymers-14-03679-f001:**
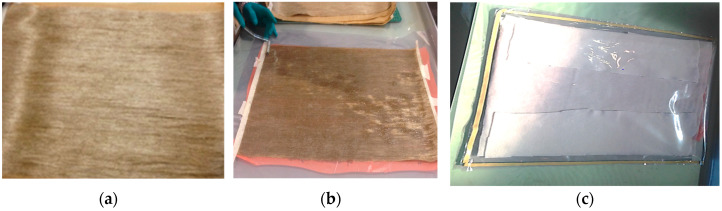
(**a**) Flax fibre sheet; (**b**) wetting fibres; (**c**) closing vacuum bagging; (**d**) vacuum bagging system; (**e**) test specimens.

**Figure 2 polymers-14-03679-f002:**
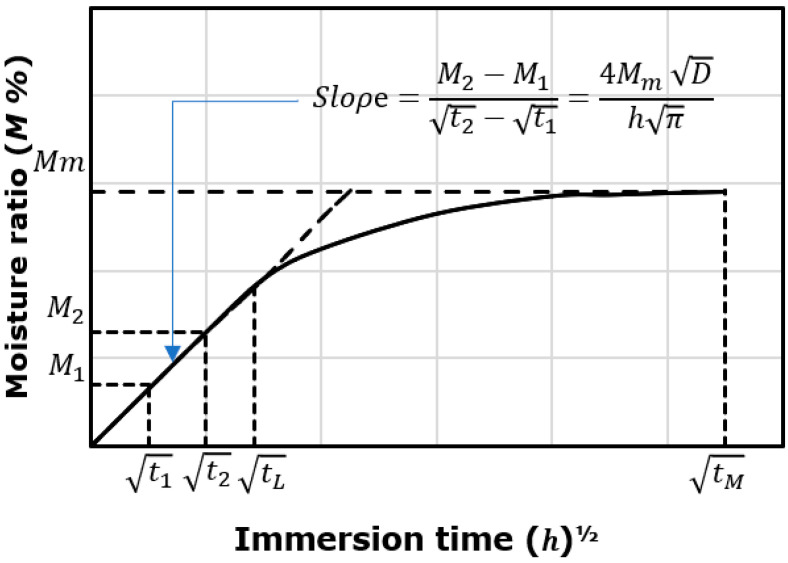
Diagram of calculating moisture content with square root of immersion time.

**Figure 3 polymers-14-03679-f003:**
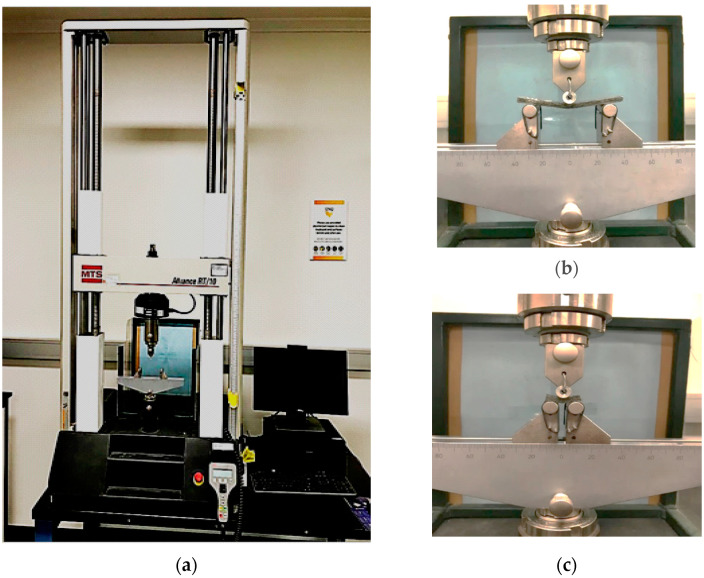
Mechanical tests of hybrid composites: (**a**) MTS machine; (**b**) flexural test and (**c**) interlaminar test.

**Figure 4 polymers-14-03679-f004:**
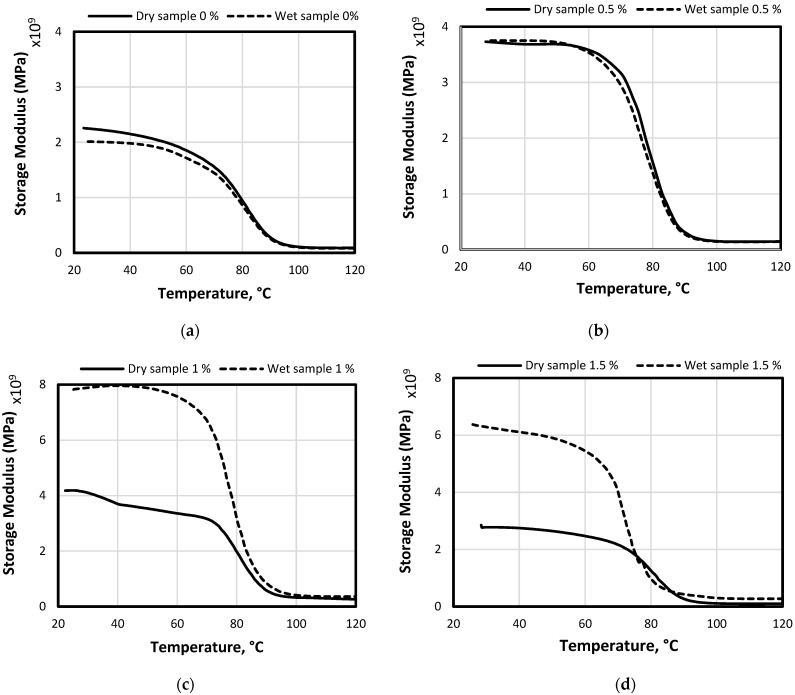
Storage modulus of dry and wet samples: (**a**) Samples with 0%; (**b**) samples with 0.5%; (**c**) samples with 1%; (**d**) samples with 1.5%.

**Figure 5 polymers-14-03679-f005:**
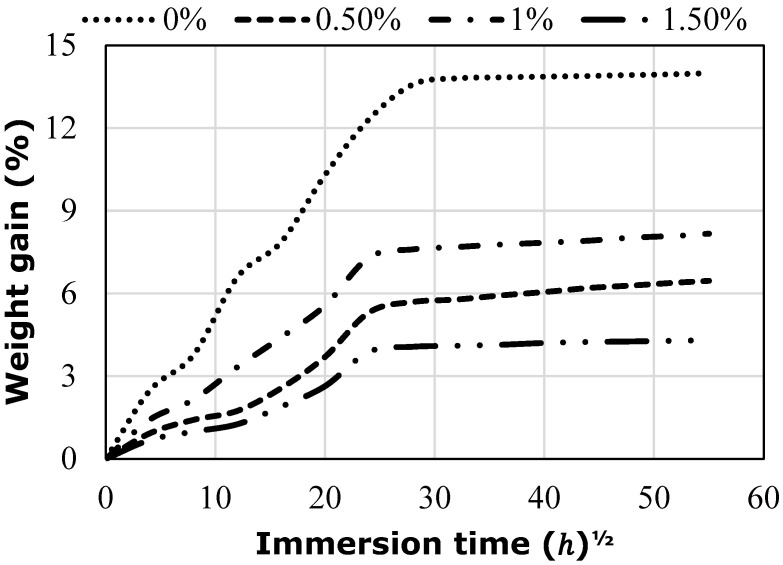
Moisture absorption behaviour of hybrid composites with various graphene weights.

**Figure 6 polymers-14-03679-f006:**
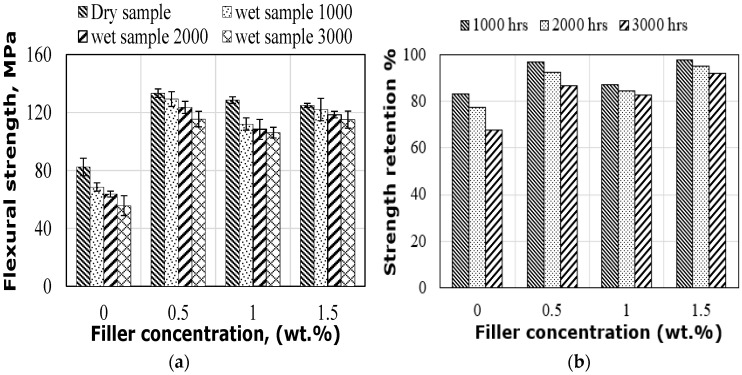
(**a**) FS under dry and wet environments; (**b**) FS retention.

**Figure 7 polymers-14-03679-f007:**
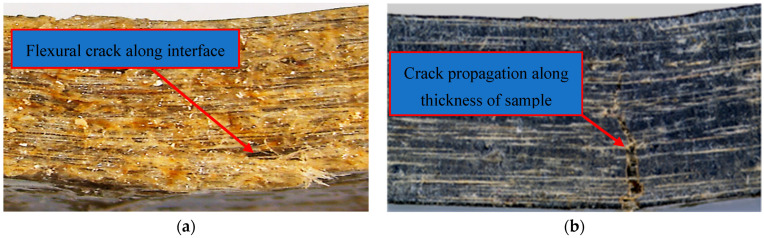
Failure of hybrid composites under flexure in wet conditions: (**a**) Unfilled composites and (**b**) filled composites.

**Figure 8 polymers-14-03679-f008:**
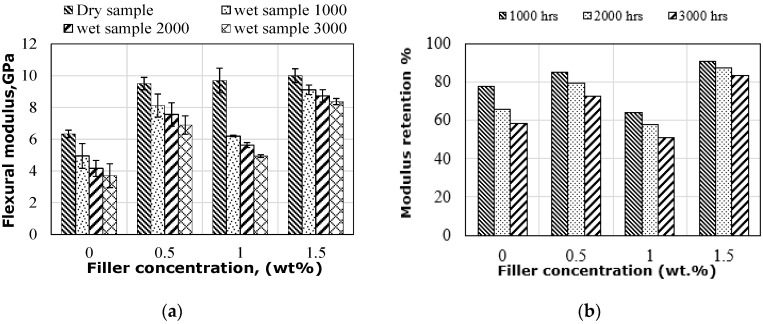
(**a**) FM under dry and wet environments and (**b**) FM retention.

**Figure 9 polymers-14-03679-f009:**
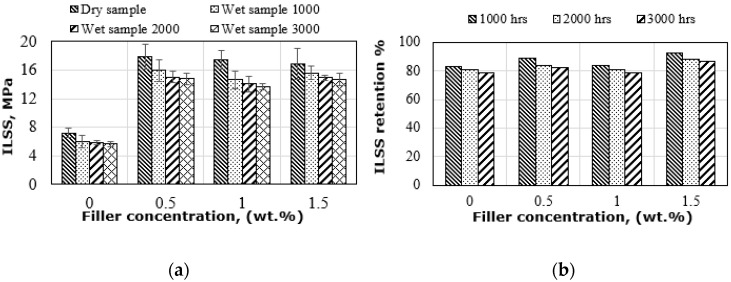
(**a**) ILSS strength under dry and wet environments and (**b**) ILSS strength retention.

**Table 1 polymers-14-03679-t001:** Internal structure of flax fibres.

Cellulose	Hemi-Cellulose	Pectin	Lignin	Wax	Moisture Ratio
62–72%	18.6–20.6%	2.3%	2–5%	1.5–1.7%	8–12%

**Table 2 polymers-14-03679-t002:** Properties of epoxy resin, natural fibres and graphene.

Material	Density (g/cm^3^)	Elastic Modulus (GPa)	Tensile Strength (MPa)
Graphene	0.03	340	130 × 10^6^
Flax fibres	1.40	70.0	1400
Epoxy resin	1.12–1.17	3.4	130

**Table 3 polymers-14-03679-t003:** Glass transition temperature of dry and wet samples.

Sample	Glass Transition Temperature		
	Dry Condition	Wet Condition	% Drop in Tg
0%	72.0 °C	71.3 °C	0.97
0.5%	76.9 °C	74.4 °C	3.25
1.0%	79.3 °C	71.0 °C	10.47
1.5%	80.3 °C	71.6 °C	10.60

**Table 4 polymers-14-03679-t004:** Average values of *M_t_%* and *D* × 10^−9^ mm^2^/s for flax/epoxy hybrid composites.

Condition	Physical Measure	Composite Materials with Filler Ratios			
		0%	0.5%	1%	1.5%
Immersion for 1000 h	*Mt%*	13.4	4.2	6.6	3.6
	*D*	4.8	2.8	3.5	1.4
Immersion for 2000 h	*Mt%*	13.6	4.4	6.8	3.8
	*D*	4.9	3.6	4.2	2.5
Immersion for 3000 h	*Mt%*	13.7	4.6	7.0	4.0
	*D*	6.0	4.5	5.4	4.4

## Data Availability

All the data required are reported in this manuscript.

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
