# Peer review of "Long-Term Water Absorption of Hybrid Flax Fibre-Reinforced Epoxy Composites with Graphene and Its Influence on Mechanical Properties"

_polymers, 2022, doi:10.3390/polym14173679_

Round 1

Reviewer 1 Report

1. Figure 1a,b,d have been published in their other paper (Polymers 2022, 14, 1841).

2. Figure 9 not showed the inter-laminar shear strength.

3. Some data such as flexural strength and inter-laminar shear strength has been published in their paper (at 20 ℃ in Polymers 2022, 14, 1841).

4. SEM results can not present the valuable information.

5. Figure 10 is too blurry. The scale bar is also missing from figures. 

6. Why choose six layers instead of odd number such as five ? 

7. Prior to adding the resin, the flax fibre sheets were placed in an oven at 40°C for 30 minutes to remove moisture as per the manufacturer's  recommendations. The dry conditions are unimaginable. What is the moisture content of flax fibre sheets before and after drying at 40℃ for 30 min?

8. The references in Table 1 and 2 are needless.

Author Response

The authors would like to thank the editor and the reviewers for their efforts and time in reviewing the paper. In the following table, the authors have attempted to respectfully answer the reviewers’ comments to enhance the quality of the paper.

Reviewer 2 Report

In this paper, the authors have investigated the effectiveness of adding graphene in flax-fiber-based composites when immersed in water. Overall, this paper is well written and the conclusions are supported by the experimental findings. However, I have a  few minor comments mentioned below that will be helpful in increasing the quality of this paper: 

1. Abstract: It will be helpful for the readers of the authors can mention the % reduction in the strength of the composite samples with exposure to moisture. 

2. What is the reason for adding graphene up to 1.5 %? Did the authors try adding more graphene? 

3. Figure 4: What is the reason for the high storage modulus of wet samples (1% and 1.5%) compared to the dry samples? 

Author Response

(The authors gave the same response as above.)

Round 2

Reviewer 1 Report

accept